# Whisper Smarter, not Harder: Adversarial Attack on Partial Suppression

## Abstract

Automatic Speech Recognition (ASR) models are deployed in an extensive range of applications. However, recent studies have demonstrated the possibility of adversarial attack on these models which could potentially suppress or disrupt model output. We investigate and verify the robustness of these attacks and explore if it is possible to increase their imperceptibility. We additionally find that by relaxing the optimisation objective from complete suppression to partial suppression, we can further decrease the imperceptibility of the attack. We also explore possible defences against these attacks and show a low-pass filter defence could potentially serve as an effective defence.

## 1 Introduction

The use of Automatic Speech Recognition (ASR) models have propagated across many platforms and technologies. These ASR models are capable of transcribing audio into text. However, these models have been shown to be susceptible to adversarial attack, which a malicious party can employ to hijack the model and generate, suppress or distort outputs from the models.

One of these ASR models is Whisper Radford et al. (2022), developed by OpenAI, which uses special start and end tokens to control its autoregressive token generation process. In particular, we focus on the end-of-sequence `<endoftext>` (EOS) token, which, when generated by Whisper, indicates that token generation should be halted and the output to be finalised.

Current attack methods on Whisper have shown that it is possible to induce a premature generation of the EOS token using a universal attack snippet trained adversarially Raina et al. (2024), but we confirm that, for smaller models, the recommended parameters can be further improved to make the attack less audible to the human ear, and thus less noticeable as an attack, without compromising majority of its attack power.

However, there are few papers to suggest that this is the only way to attack Whisper. We posit that if the criteria for attack is less strict, the attack is counter-intuitively better at disrupting output. Specifically, if we impose a generation leniency instead of a hard limit of one token, we can potentially create a more robust attack.

## 2 Setup

### 2.1 Model

Whisper consists of a typical encoder-decoder transformer architecture. During transcription, Whisper first converts input audio into its log Mel spectrogram form via Short Fast Fourier Transform performed over windowed segments in 30-second chunks. Its encoder then accepts this log Mel spectrogram and outputs audio features, which the decoder uses to autoregressively generate tokens until the EOS token is generated.

## 2.2 Data

The data used is extracted from the TED-LIUM corpus, release 1 Rousseau et al. (2012), consisting "train", "validation" and "test" splits. Snippets are trained on the "train" split, validated on the "validation" split, and evaluated on the "test" split.

## 2.3 Methodology

In the following experiments, we compare the results from the `small.en` ("small") and `tiny.en` ("tiny") models of Whisper, as well as the baseline performance on the same sizes of models ("base small" and "base tiny") using random attack snippets. All training hyperparameters used can be found in Appendix A.1 and details of the training setup and hardware in Appendix A.2.

# 3 Complete Suppression

## 3.1 Overview

Current attack methods recommend creating a universal attack snippet with gradient-based training, where the snippet magnitude is clamped to 0.02, length set at 0.64s (corresponding to 10,240 frames of audio for a 16 kHz sampling rate) and prepended to the audio to attack on. We verify this with experiments varying the clamp limit $\varepsilon$, snippet length $L$ and position of prepend $T$. Full experimental data is given in Appendices B.1.1, B.1.2 and B.1.3 respectively.

## 3.2 Optimisation Objective

We borrow the optimisation objective from the current attack method for total suppression Raina et al. (2024):

$$\mathbf{a} = \arg\max_{\mathbf{a}} \left\{ \prod_{i=1}^{D} P(y_1 = \texttt{eos}|\mathbf{a} \oplus \mathbf{x}^{(i)}, \mathbf{y}_0^*) \right\}$$

$$= \arg\max_{\mathbf{a}} \left\{ \sum_{i=1}^{D} \log P(y_1 = \texttt{eos}|\mathbf{a} \oplus \mathbf{x}^{(i)}, \mathbf{y}_0^*) \right\}$$

where $\mathbf{a}$ is the attack snippet, $\mathbf{x}^{(i)}$ is the audio from dataset $D$ to be attacked, $\mathbf{y}_0^* = $ `<|startoftranscript|>` `<lang tag><|task tag|>` is the starting series of tokens that initiates the token generation, and $y_1$ is the first token to be properly generated. This objective essentially seeks to maximise the log probability of generating the EOS token at the first token position.

## 3.3 Metrics

We measure the effectiveness of attacks for complete suppression in terms of the rate of empty transcriptions $\varnothing$ and the average sequence length (ASL):

$$\varnothing = \frac{1}{D} \sum_{i}^{D} \mathbf{1} \left\{ \tilde{y}_1^{*(i)} = \texttt{eot} \right\}$$

$$\text{ASL} = \frac{1}{D} \sum_{i}^{D} \texttt{len} \left( \tilde{\mathbf{y}}^{*(i)} \right)$$

where $\tilde{y}_1^{*(i)}$ is the first generated token and $\texttt{len}(\cdot)$ is the length of the transcription given. Consequently, a higher $\varnothing$ or a lower ASL represents stronger suppressive attack power.

### 3.4 Varying Clamp Limits $\varepsilon$

We define the "clamp limit" $\varepsilon$ as the magnitude in which the values in the attack tensor are clamped to, i.e. $\|\mathbf{a}\|_\infty \leq \varepsilon$, where $\|\cdot\|_\infty$ is the l-infinity norm. The attack length and positions are kept constant at $L = 0.64s$ and $T = 0s$ respectively.

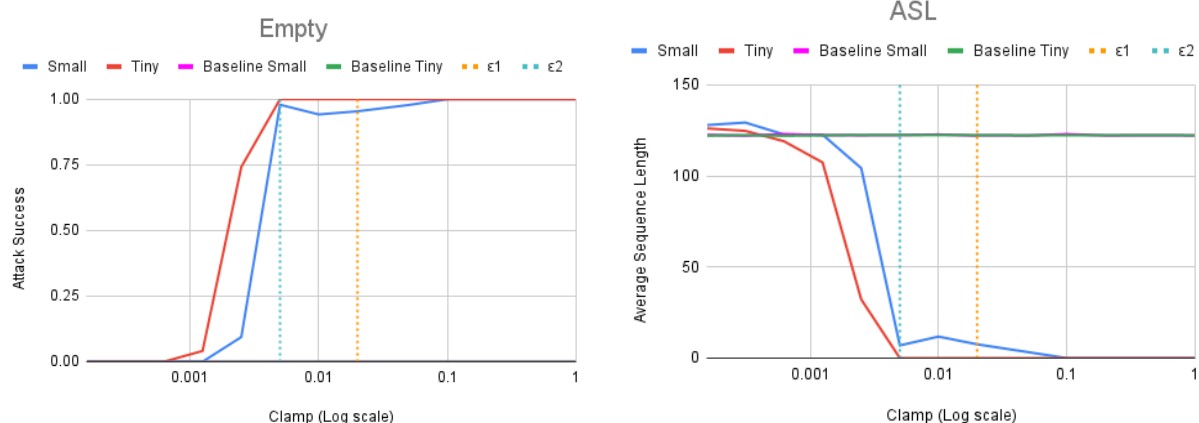

Figure 1: The $\varnothing$ and ASL graphs over various clamp values on the logarithmic scale, with $\varepsilon_1$ and $\varepsilon_2$ notated as orange and blue dotted lines respectively.

The graphs given in Fig.1 verify that, for the smaller models, despite the recommended clamp limit of 0.02 ($\varepsilon_1$), clamp limits that are much smaller are still effective up until approximately 0.005 ($\varepsilon_2$).

### 3.5 Varying Length $L$

We define the length $L$ as the length of the snippet in seconds. In varying the length of the snippet, we keep the clamp limit constant at $\varepsilon = 0.005$ and position at $T = 0s$.

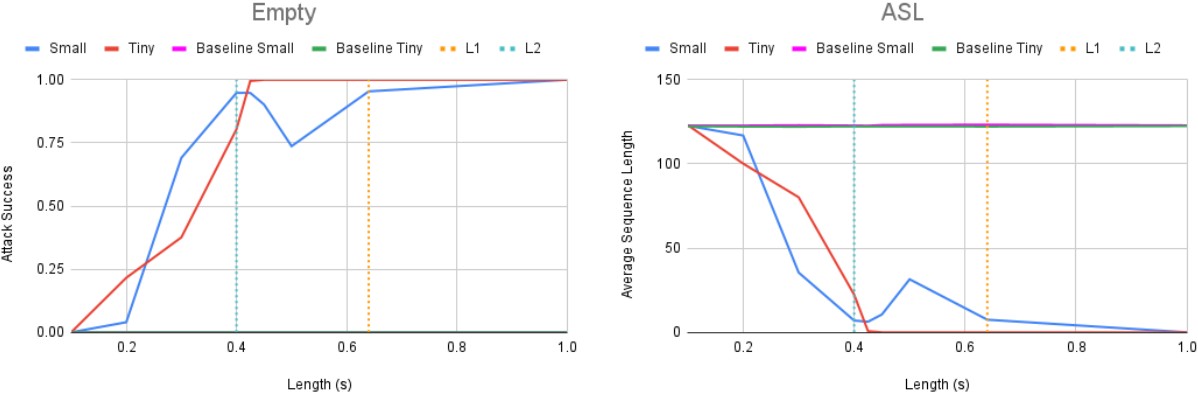

Figure 2: The $\varnothing$ and ASL graphs over various lengths, with $L_1$ and $L_2$ notated as orange and blue dotted lines respectively.

Again, the graphs in Fig.2 show potential improvement for both smaller models. Current recommended attack lengths of $L_1 = 0.64s$ in orange can still be decreased to around $L_2 = 0.4s$.

### 3.6 Varying Position $T$

We define this as the time position (in seconds) at which to insert the snippet. Due to the various audio lengths in the train dataset $D$, we have decided to impose a constraint $0 \leq T \leq 1$ to minimise silences occur after the audio and before the snippet. In varying the position of the snippet, we keep the clamp limit constant at $\varepsilon = 0.005$ and length at $L = 0.4s$.

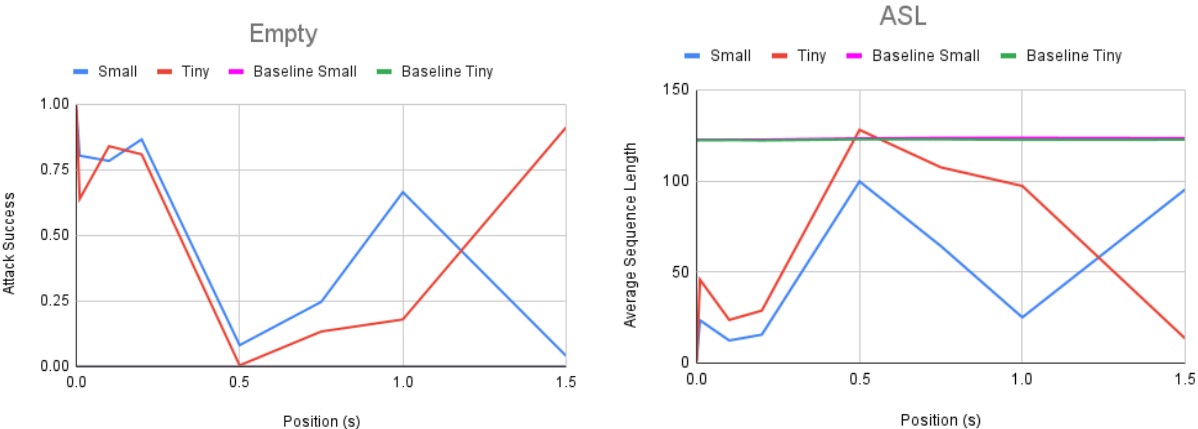

Figure 3: The $\varnothing$ and ASL graphs over various positions of prepend.

Contrary to earlier findings, Fig.3 shows that despite a potential improvement to around $T = 0.2s$, prepend at $T = 0s$ is still the most effective way to attack.

## 4 Partial Suppression

### 4.1 Overview

Instead of focusing on complete suppression of output, we try to relax the optimisation objective to allow some, but not complete, token generation. In doing so, we determine if the attack can be made even more imperceptible in terms of clamp limits $\varepsilon$ and length $L$. Because position has been shown to be best at $T = 0s$, we will not explore varying positions for partial suppression. Full experimental data is given in Appendices B.2.1 and B.2.2 respectively.

### 4.2 Optimisation Objective

Since the goal is similar to that of fine-tuning in Large Language Models, we can use a similar optimisation objective. The objective in fine-tuning is given Qi et al. (2024):

$$\min_{\theta} \left\{ \mathbb{E}_{(\boldsymbol{x}, \boldsymbol{y}) \sim D} - \sum_{t=1}^{|\boldsymbol{y}|} \log \pi_{\theta}(y_t | \boldsymbol{x}, \boldsymbol{y}_{<t}) \right\}$$

where $\pi_{\theta}$ is defined as the language model $\pi$ with parameters $\theta$. The objective seeks to minimise the "per-token cross-entropy loss at each token position $t$" Qi et al. (2024).

We can produce a modified objective for partial suppression:

$$\mathbf{a} = \arg\min_{\mathbf{a}} \left\{ \mathbb{E}_{(\mathbf{x},t)\sim D} - \sum_{t=1}^{\min(\delta,|\mathbf{y}|)} \log P(y_t = \texttt{eos}|\mathbf{a} \oplus \mathbf{x}, \mathbf{y}_{<t}) \right\}$$

$$= \arg\max_{\mathbf{a}} \left\{ \sum_{i=1}^{D} \frac{1}{T} \sum_{t=1}^{T} \log P(y_t^{(i)} = \texttt{eos}|\mathbf{a} \oplus \mathbf{x}^{(i)}, \mathbf{y}_{<t}^{(i)}) \right\}$$

where $T = \min(\delta, |\mathbf{y}^{(i)}|)$. That is, for all examples, we wish to maximise the probability of the EOS token generated within a certain number of tokens $\delta$ from the start.

### 4.3 Metrics

Since $\varnothing$ and ASL are better indicators of complete suppression rather than disruptiveness, we introduce the average BiLingual Evaluation Understudy (BLEU) score $\bar{B}$:

$$\bar{B} = \frac{1}{D} \sum_{i=1}^{D} BLEU' \left( \tilde{\mathbf{y}}^{*(i)}, \mathbf{Y}^{(i)} \right)$$

$$BLEU' \left( \tilde{\mathbf{y}}^{*(i)}, \mathbf{Y}^{(i)} \right) = \begin{cases} 0, & \text{if } |\mathbf{y}^{*(i)}| = 0 \\ BLEU \left( \tilde{\mathbf{y}}^{*(i)}, \mathbf{Y}^{(i)} \right), & \text{otherwise} \end{cases}$$

where $\mathbf{Y}^{(i)}$ is the ground truth for that example and $BLEU(\cdot, \cdot)$ is the BLEU score of the transcription against its ground truth, and the average Word Error Rate (WER) $\bar{W}$:

$$\bar{W} = \frac{1}{D} \sum_{i=1}^{D} WER \left( \tilde{\mathbf{y}}^{*(i)}, \mathbf{Y}^{(i)} \right)$$

where $WER(\cdot, \cdot)$ is the WER score of the transcription against its ground truth. A lower BLEU score or a higher WER represents stronger attack power.

### 4.4 Varying Clamp Limits $\varepsilon$

Like the experiments for complete suppression, we evaluate the clamp limit with $L = 0.64s$ and prepend at $T = 0s$.

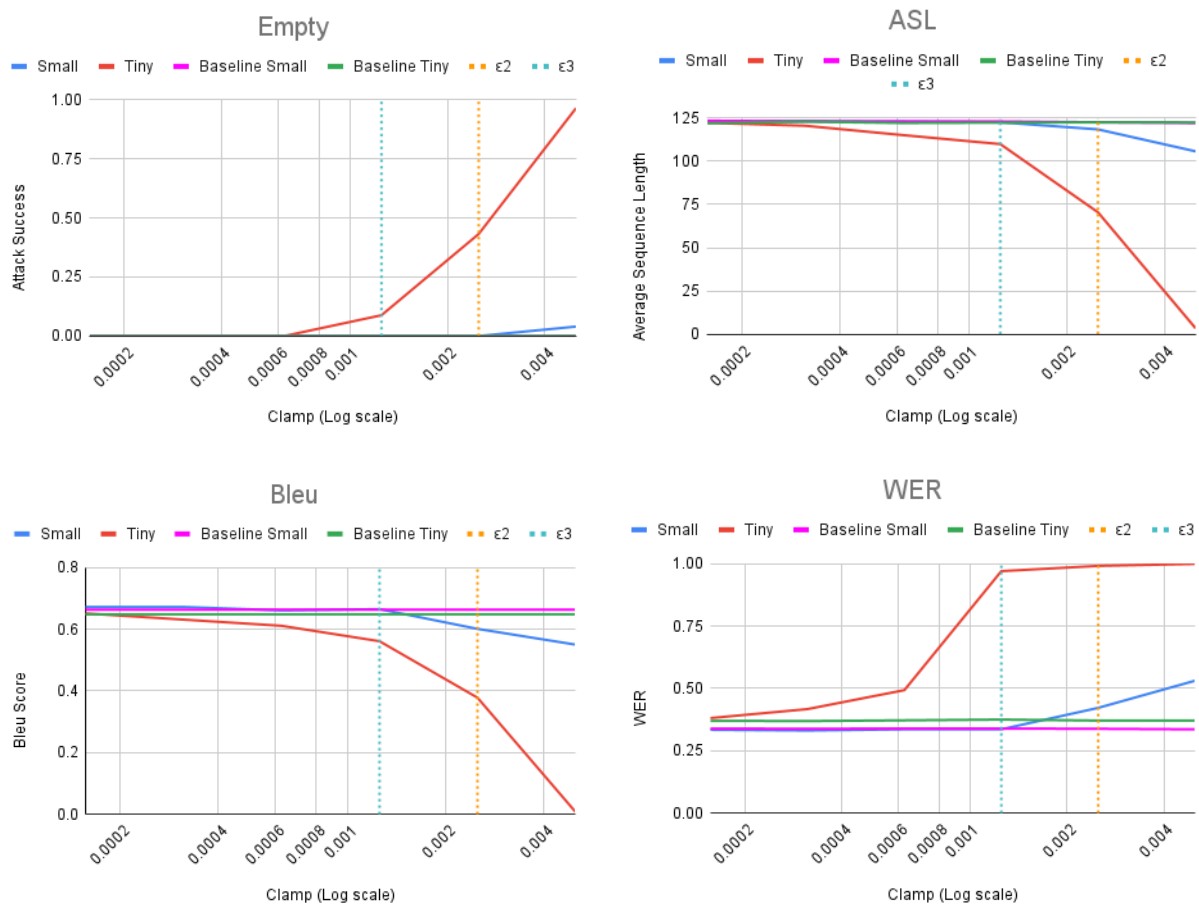

Figure 4: The ∅, ASL, BLEU score and WER graphs over various clamp limits, with $\varepsilon_2$ and $\varepsilon_3$ notated as orange and blue dotted lines respectively.

Despite the poorer ∅ and ASL scores, the WER graph in Fig.4 suggests a possible further decrease in clamp limits, from around $\varepsilon_2 = 0.0025$ to $\varepsilon_3 = 0.00125$.

### 4.5 Varying Length $L$

Similarly, we set $\varepsilon = 0.005$ and prepend at $T = 0s$.

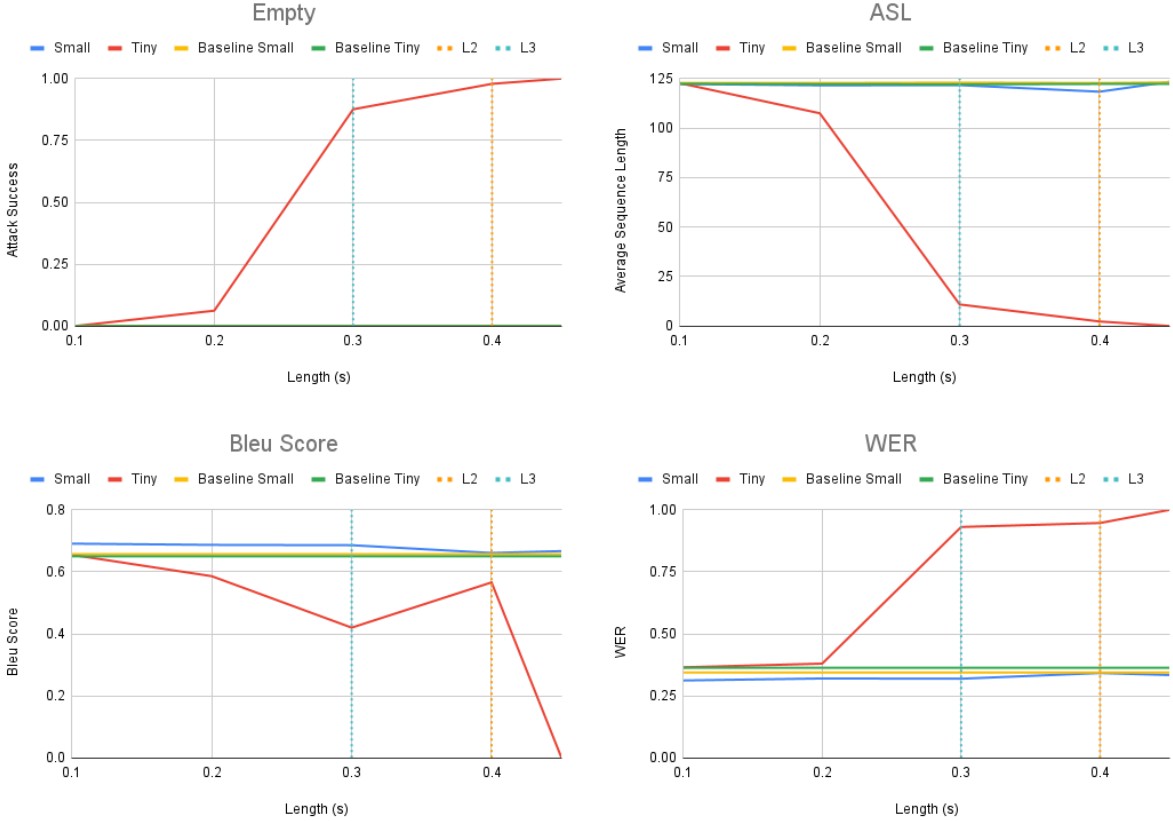

Figure 5: The $\varnothing$, ASL, BLEU score and WER graphs over various lengths, with $L_2$ and $L_3$ notated as orange and blue dotted lines respectively.

Likewise, Fig.5 shows that the attack snippet length can be lowered further, from around $L_2 = 0.4s$ to $L_3 = 0.3s$.

## 5 Transferability

Here, we briefly explore the transferability of both attacks across model sizes to affirm findings from previous works and determine if the proposed attack on partial suppression is better in this regard. We train an attack on a surrogate model and evaluate the metrics on a victim model. Once again, the two model sizes tested will be `small.en` and `tiny.en`, $\varepsilon = 0.005$, $L = 10,240$ and $T = 0$.

Table 1: Table of Metrics for transferability evaluation.

| Surrogate | | small.en | | tiny.en | |
|---|---|---|---|---|---|
| **Victim** | | small.en | tiny.en | small.en | tiny.en |
| No Attack | ∅ | 0 | | 0 | |
| | ASL | 123 | | 122.8 | |
| | WER | 0.344 | | 0.363 | |
| Complete | ∅ | 0.948 | 0 | 0 | 0.804 |
| | ASL | 7 | 122.4 | 122.7 | 22.4 |
| | WER | 1 | 0.353 | 0.333 | 1 |
| Partial | ∅ | 0.225 | 0 | 0 | 0.979 |
| | ASL | 69 | 122.7 | 123.8 | 2.3 |
| | WER | 0.665 | 0.349 | 0.362 | 0.999 |

Note that when no attack is applied, attack performance is independent on the surrogate model. The results from Table 1 seem to indicate that both attacks have similar non-transferability across model sizes, i.e. attacks trained on the surrogate model perform poorly on a different victim model.

## 6 Defences

### 6.1 Overview

In order to counter the gradient-based adversarial attacks, a few simple audio defences were tested and benchmarked. The defences explored in particular were the Butterworth Low-pass filter, Mu Law compression and Mu Law compression with decompression. All experiments here are run on clamp $\varepsilon = 0.005$, length $L = 0.64s$ and prepended at $T = 0s$. The model used is `tiny.en`, and the attacking snippet was trained on complete suppression. All experimental data is given under Appendices B.3.1 and B.3.2 for the Mu-Law algorithms and Low-pass filter respectively.

### 6.2 Metrics

In order to measure defensive power, we first record the attack metrics ($\varnothing$, ASL, Bleu score and WER) with and without the attack snippet, then find the difference between these values. This is the baseline attack power $\alpha_{base}$:

$$\alpha_{base} = \sum_{i=1}^{D} M\left(\mathbf{a} \oplus \mathbf{x}^{(i)}\right) - M\left(\mathbf{x}^{(i)}\right)$$

where $M(\cdot)$ refers to the attack metrics. We then measure the the differences in attack metrics with the defence $\alpha_d$:

$$\alpha_d = \sum_{i=1}^{D} M\left(d(\mathbf{a} \oplus \mathbf{x}^{(i)})\right) - M\left(d(\mathbf{x}^{(i)})\right)$$

where $d(\cdot)$ is the applied defence. Finally, we calculate:

$$\alpha_{\%} = \frac{\alpha_d}{\alpha_{base}} \times 100\%$$

to find the percentage of retained attack power after defence. The smaller $\alpha_{\%}$ is, the smaller the retained attack power and the more powerful the defence.

### 6.3 Mu Law

Mu-Law compression is often used in telecommunications and is a form of non-uniform quantization. Here, we tested both Mu-Law compression only ($\mu(x)$) and Mu-Law compression with decompression ($\mu'(\mu(x))$),

whose equations are given:

$$\mu(x) = sign(x)\frac{\ln(1 + \mu\,|x|)}{\ln(1 + \mu)}$$

$$\mu'(x) = sign(x)\frac{(\mu + 1)^{|x|} - 1}{\mu}$$

where $\mu$ is the non-linearity of the compression. Here, it is set to $\mu = 255$ to simulate a typical 16-bit to 8-bit quantization.

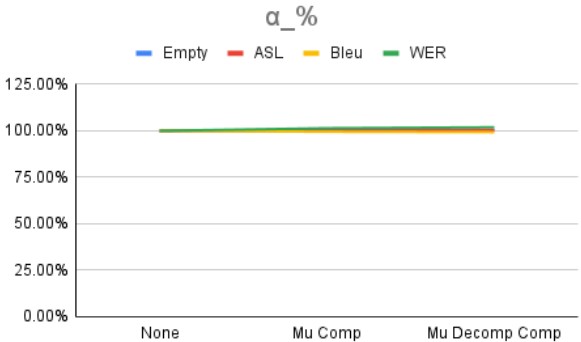

Figure 6: $\alpha_\%$ over no compression ("None"), Mu-Law compression ("Mu Comp") and Mu-Law compression with decompression ("Mu Decomp Comp").

It is clear from Fig.6 that both compression algorithms fail to decrease attack power at all since $\alpha_\%$ remains constant throughout. This proves that the Mu Law algorithms are ineffective defences.

### 6.4 Butterworth Low-pass Filter

The equation in frequency $f$ for the Butterworth Low-pass filter is given:

$$LP(f) = |H(f)|\,f = \frac{1}{\sqrt{1 + \left(\frac{f}{f_{cutoff}}\right)^{2n}}} \times f$$

where $f$ is the input frequency, $f_{cutoff}$ is the cut-off frequency for the filter and $n$ is the order of the filter, i.e. how harsh the taper at the cut-off frequency is. The order was set to $n = 5$ and different cut-off frequencies $f_{cutoff}$ were tested.

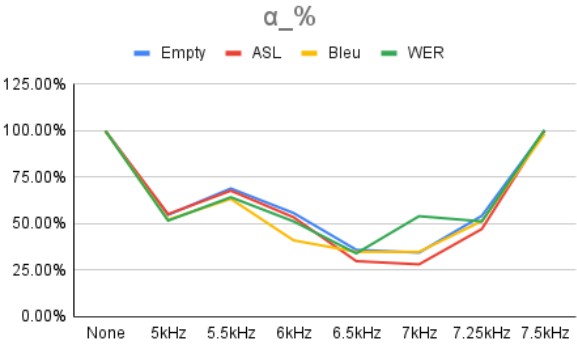

Figure 7: $\alpha_\%$ over a range of cut-off frequencies.

Interestingly, $\alpha_\%$ seems to be lowest in the range $6.5\text{kHz} \leq f_{cutoff} \leq 7\text{kHz}$, suggesting a potential correlation between attack power and frequency strengths in the range. It is possible that the low-pass filter, if configured correctly and evaluated against a broader range of attacks, could be a potential robust defence against such gradient-based attacks.

## 7 Limitations

Whilst the results produced are conclusive, they are only limited to two of Whisper's smallest model sizes, which are less robust than bigger sizes like `large.en` or `turbo`. Further testing is needed to ensure the results are applicable and generalisable to larger model sizes and even other ASR models.

Secondly, due to time and resource constraints, the experiments were carried out on a specific subset of one audio dataset. It is yet to be shown that the results are generalisable to most examples on most datasets.

Next, most results were produced in one pass only, thus it is impossible to differentiate which values are outliers and which are not. This could lead to false positives and trends that may not accurately reflect the true weaknesses and strengths of the attack or the model.

It is also worth mentioning that experiments for the defences on an attacker trained on partial suppression have not be carried out, meaning both attacks cannot be accurately assessed and compared based on robustness and defence evasion.

Lastly, the results for the attack on partial suppression, specifically `small.en`, may not be representative as, due to resource constraints, the model was unable to converge.

## 8 Conclusion

In summary, we affirm that current gradient-based attack methods for smaller models can still be improved and made more imperceptible for smaller models. Furthermore, we provide promising evidence that the attack on partial suppression is better than the traditional attack on complete suppression. Lastly, we show that basic audio defences like low-pass filters seem to possess some defensive power against the traditional gradient-based attack on complete suppression and thus can form the basis of future defensive methods.

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

# A  Training Details

## A.1  Training Hyperparameters

The optimizer used is the Adaptive Moment Estimation with Weight Decay Regularization, or AdamW Raina et al. (2024), a variation of the Adam algorithm. The learning rate is set to 1e-3. The training loop employs early stopping with a patience of 5, a time limit of 45 minutes and an iteration limit of 30.

In order to better account for convergence, we also implement a requirement for a minimum decrease in validation loss, that is, patience decreases if the validation loss increases or if it decreases by less than a certain amount.

## A.2  Data and Hardware

Training and testing were carried out on an Nvidia A40 GPU with 48GBs of GPU memory.

All experiments were executed with 500 training examples, 150 validation examples and 250 test examples. The reason for the choice of small dataset sizes is due to the high training duration and low time constraint. Train and validation data were loaded onto PyTorch DataLoaders with batch size 1.

# B  Experimental Results

## B.1  Complete Suppression

### B.1.1  Clamp Limit $\varepsilon$

| | $\varepsilon$ | 0.00015625 | 0.0003125 | 0.000625 | 0.00125 | 0.0025 | 0.005 | 0.005 | 0.01 | 0.02 | 0.02 | 0.05 | 0.1 | 0.2 | 0.5 | 1 |
|---|---|---|---|---|---|---|---|---|---|---|---|---|---|---|---|---|
| Empty | Small | 0 | 0 | 0 | 0 | 0.094 | 0.979 | 0.979 | 0.942 | 0.954 | 0.954 | 0.978 | 1 | 1 | 1 | 1 |
| | Tiny | 0 | 0 | 0 | 0.041 | 0.742 | 1 | 1 | 1 | 1 | 1 | 1 | 1 | 1 | 1 | 1 |
| | Baseline Small | 0 | 0 | 0 | 0 | 0 | 0 | 0 | 0 | 0 | 0 | 0 | 0 | 0 | 0 | 0 |
| | Baseline Tiny | 0 | 0 | 0 | 0 | 0 | 0 | 0 | 0 | 0 | 0 | 0 | 0 | 0 | 0 | 0 |
| ASL | Small | 127.857 | 129.238 | 122.716 | 122.596 | 104.23 | 6.964 | 6.964 | 11.801 | 7.572 | 7.572 | 3.261 | 0 | 0 | 0 | 0 |
| | Tiny | 126.039 | 124.675 | 118.985 | 107.348 | 32.206 | 0 | 0 | 0 | 0 | 0 | 0 | 0 | 0 | 0 | 0 |
| | Baseline Small | 122.5 | 122.1 | 122.9 | 122.4 | 122.3 | 122.4 | 122.4 | 122.7 | 122.1 | 122.3 | 122.2 | 122.9 | 122.2 | 122.3 | 122.2 |
| | Baseline Tiny | 122.1 | 122.3 | 122 | 122.2 | 122.4 | 122.4 | 122.4 | 122.4 | 122.3 | 122.3 | 122.2 | 122.3 | 122.3 | 122.3 | 122.2 |

### B.1.2  Length $L$

| | Length (s) | 0.1 | 0.2 | 0.3 | 0.4 | 0.4 | 0.425 | 0.45 | 0.5 | 0.64 | 0.64 | 1 |
|---|---|---|---|---|---|---|---|---|---|---|---|---|
| Empty | Small | 0 | 0.04 | 0.691 | 0.948 | 0.948 | 0.948 | 0.902 | 0.737 | 0.954 | 0.954 | 1 |
| | Tiny | 0 | 0.216 | 0.376 | 0.804 | 0.804 | 0.995 | 1 | 1 | 1 | 1 | 1 |
| | Baseline Small | 0 | 0 | 0 | 0 | 0 | 0 | 0 | 0 | 0 | 0 | 0 |
| | Baseline Tiny | 0 | 0 | 0 | 0 | 0 | 0 | 0 | 0 | 0 | 0 | 0 |
| ASL | Small | 122.7 | 116.9 | 35.5 | 7 | 7 | 6.3 | 10.6 | 31.5 | 7.5 | 7.5 | 0 |
| | Tiny | 123.1 | 100.2 | 80.2 | 22.4 | 22.4 | 0.7 | 0 | 0 | 0 | 0 | 0 |
| | Baseline Small | 122.8 | 122.8 | 123 | 122.8 | 122.8 | 122.7 | 123.1 | 123.2 | 123.3 | 123.3 | 122.9 |
| | Baseline Tiny | 122.4 | 122.2 | 122.1 | 122.3 | 122.3 | 122.3 | 122.3 | 122.3 | 122.2 | 122.2 | 122.5 |

### B.1.3 Position $T$

| Position (s) | | 0 | 0.01 | 0.1 | 0.2 | 0.5 | 0.75 | 1 | 1.5 |
|---|---|---|---|---|---|---|---|---|---|
| Empty | Small | 0.979 | 0.804 | 0.784 | 0.866 | 0.082 | 0.247 | 0.665 | 0.041 |
| | Tiny | 0.995 | 0.639 | 0.84 | 0.809 | 0.005 | 0.134 | 0.18 | 0.912 |
| | Baseline Small | 0 | 0 | 0 | 0 | 0 | 0 | 0 | 0 |
| | Baseline Tiny | 0 | 0 | 0 | 0 | 0 | 0 | 0 | 0 |
| ASL | Small | 2.042 | 23.5 | 12.4 | 15.6 | 99.7 | 64.2 | 25.07 | 95.3 |
| | Tiny | 0.7 | 45.8 | 23.7 | 28.8 | 128 | 107.4 | 97.2 | 13.5 |
| | Baseline Small | 122.7 | 122.5 | 122.6 | 122.6 | 123.3 | 123.6 | 123.6 | 123.4 |
| | Baseline Tiny | 122.1 | 122.4 | 122.4 | 122.2 | 122.8 | 122.8 | 122.5 | 122.6 |

## B.2 Partial Suppression

### B.2.1 Clamp limits $\varepsilon$

| $\varepsilon$ | | 0.00015625 | 0.0003125 | 0.000625 | 0.00125 | 0.00125 | 0.0025 | 0.0025 | 0.005 |
|---|---|---|---|---|---|---|---|---|---|
| Empty | Small | 0 | 0 | 0 | 0 | 0 | 0 | 0 | 0.04 |
| | Tiny | 0 | 0 | 0 | 0.088 | 0.088 | 0.432 | 0.432 | 0.964 |
| | Baseline Small | 0 | 0 | 0 | 0 | 0 | 0 | 0 | 0 |
| | Baseline Tiny | 0 | 0 | 0 | 0 | 0 | 0 | 0 | 0 |
| ASL | Small | 122.8 | 123.1 | 122.9 | 122.5 | 122.5 | 118.4 | 118.4 | 105.7 |
| | Tiny | 122.2 | 120.5 | 115.1 | 109.9 | 109.9 | 70.5 | 70.5 | 3.46 |
| | Baseline Small | 123.4 | 123.1 | 123 | 123 | 123 | 122.5 | 122.5 | 122.1 |
| | Baseline Tiny | 122 | 122.9 | 122.1 | 122.3 | 122.3 | 122.5 | 122.5 | 122.4 |
| BLEU | Small | 0.672 | 0.672 | 0.661 | 0.665 | 0.665 | 0.601 | 0.601 | 0.55 |
| | Tiny | 0.651 | 0.631 | 0.611 | 0.561 | 0.561 | 0.378 | 0.378 | 0.008 |
| | Baseline Small | 0.663 | 0.663 | 0.663 | 0.663 | 0.663 | 0.663 | 0.663 | 0.663 |
| | Baseline Tiny | 0.648 | 0.648 | 0.648 | 0.648 | 0.648 | 0.648 | 0.648 | 0.648 |
| WER | Small | 0.334 | 0.331 | 0.336 | 0.336 | 0.336 | 0.422 | 0.422 | 0.531 |
| | Tiny | 0.381 | 0.417 | 0.493 | 0.97 | 0.97 | 0.991 | 0.991 | 0.999 |
| | Baseline Small | 0.339 | 0.338 | 0.339 | 0.339 | 0.339 | 0.338 | 0.338 | 0.336 |
| | Baseline Tiny | 0.37 | 0.369 | 0.372 | 0.375 | 0.375 | 0.371 | 0.371 | 0.371 |

### B.2.2 Length $L$

| Length (s) | | 0.1 | 0.2 | 0.3 | 0.3 | 0.4 | 0.4 | 0.45 |
|---|---|---|---|---|---|---|---|---|
| Empty | Small | 0 | 0 | 0 | 0 | 0 | 0 | 0 |
| | Tiny | 0 | 0.062 | 0.876 | 0.876 | 0.979 | 0.979 | 1 |
| | Baseline Small | 0 | 0 | 0 | 0 | 0 | 0 | 0 |
| | Baseline Tiny | 0 | 0 | 0 | 0 | 0 | 0 | 0 |
| ASL | Small | 122.3 | 121.6 | 121.7 | 121.7 | 118.4 | 118.4 | 123.4 |
| | Tiny | 122.7 | 107.5 | 10.9 | 10.9 | 2.3 | 2.3 | 0 |
| | Baseline Small | 122.8 | 122.8 | 123 | 123 | 122.8 | 122.8 | 123.1 |
| | Baseline Tiny | 122.4 | 122.2 | 122.1 | 122.1 | 122.3 | 122.3 | 122.3 |
| Bleu | Small | 0.691 | 0.687 | 0.686 | 0.686 | 0.661 | 0.661 | 0.667 |
| | Tiny | 0.655 | 0.586 | 0.42 | 0.42 | 0.566 | 0.566 | 0 |
| | Baseline Small | 0.657 | 0.657 | 0.657 | 0.657 | 0.657 | 0.657 | 0.657 |
| | Baseline Tiny | 0.65 | 0.65 | 0.65 | 0.65 | 0.65 | 0.65 | 0.65 |
| WER | Small | 0.312 | 0.32 | 0.319 | 0.319 | 0.342 | 0.342 | 0.334 |
| | Tiny | 0.365 | 0.38 | 0.931 | 0.931 | 0.947 | 0.947 | 1 |
| | Baseline Small | 0.344 | 0.344 | 0.344 | 0.344 | 0.344 | 0.344 | 0.344 |
| | Baseline Tiny | 0.363 | 0.363 | 0.363 | 0.363 | 0.363 | 0.363 | 0.363 |

## B.3 Defence

### B.3.1 Mu-Law Algorithms

|  | Defence | None | Mu Comp | Mu Decomp Comp |
|---|---|---|---|---|
| Attacked | Empty | 1 | 1 | 1 |
| | ASL | 0 | 0 | 0 |
| | Bleu | 0 | 0 | 0 |
| | WER | 1 | 1 | 1 |
| Unattacked | Empty | 0.015 | 0.01 | 0.01 |
| | ASL | 125.76 | 125.9 | 126.2 |
| | Bleu | 0.435 | 0.433 | 0.432 |
| | WER | 0.569 | 0.563 | 0.561 |
| $\alpha$ | Empty | 0.985 | 0.99 | 0.99 |
| | ASL | -125.76 | -125.9 | -126.2 |
| | Bleu | -0.435 | -0.433 | -0.432 |
| | WER | 0.431 | 0.437 | 0.439 |
| $\alpha_\%$ | Empty | 100 | 100.5 | 100.5 |
| | ASL | 100 | 100.1 | 100.3 |
| | Bleu | 100 | 99.5 | 99.3 |
| | WER | 100 | 101.4 | 101.9 |

### B.3.2 Low-pass Filter

|  | Critical Frequency | None | 5kHz | 5.5kHz | 6kHz | 6.5kHz | 7kHz | 7.25kHz | 7.5kHz |
|---|---|---|---|---|---|---|---|---|---|
| With Snippet | Empty | 1 | 0.55 | 0.685 | 0.555 | 0.37 | 0.35 | 0.55 | 0.995 |
| | ASL | 0 | 56.2 | 40.39 | 58.4 | 87.9 | 91.3 | 65.9 | 0.935 |
| | Bleu | 0 | 0.202 | 0.152 | 0.253 | 0.278 | 0.282 | 0.205 | 0.003 |
| | WER | 1 | 0.786 | 0.846 | 0.79 | 0.717 | 0.795 | 0.786 | 0.997 |
| Without Snippet | Empty | 0.015 | 0.01 | 0.005 | 0.005 | 0.015 | 0.01 | 0.015 | 0.01 |
| | ASL | 125.76 | 125.6 | 125.7 | 125.7 | 125.5 | 126.8 | 125.3 | 127 |
| | Bleu | 0.435 | 0.429 | 0.428 | 0.432 | 0.43 | 0.434 | 0.429 | 0.431 |
| | WER | 0.569 | 0.563 | 0.569 | 0.569 | 0.57 | 0.562 | 0.565 | 0.564 |
| Alpha | Empty | 0.985 | 0.54 | 0.68 | 0.55 | 0.355 | 0.34 | 0.535 | 0.985 |
| | ASL | -125.76 | -69.4 | -85.31 | -67.3 | -37.6 | -35.5 | -59.4 | -126.065 |
| | Bleu | -0.435 | -0.227 | -0.276 | -0.179 | -0.152 | -0.152 | -0.224 | -0.428 |
| | WER | 0.431 | 0.223 | 0.277 | 0.221 | 0.147 | 0.233 | 0.221 | 0.433 |
| $\alpha_\%$ | Empty | 100 | 54.8 | 69 | 55.8 | 36 | 34.5 | 54.3 | 100 |
| | ASL | 100 | 55.2 | 67.8 | 53.5 | 29.9 | 28.2 | 47.2 | 100.2 |
| | Bleu | 100 | 52.2 | 63.4 | 41.1 | 34.9 | 34.9 | 51.1 | 98.4 |
| | WER | 100 | 51.7 | 064.3 | 51.3 | 34.1 | 54.1 | 51.3 | 100.5 |

