# OpenReview forum: "Whisper Smarter, not Harder: Adversarial Attack on Partial Suppression"
_TMLR — Withdrawn by Authors_

### Review · Reviewer_G6uN · 2025-08-16

**Summary Of Contributions:**

This paper studies adversarial attacks on OpenAI’s Whisper ASR models, focusing on the end-of-sequence (EOS) token attack described in Raina et al. (2024). The main contributions are:
- A hyperparameter sweep over clamp limits, snippet length, and insertion position for complete suppression attacks, showing that more imperceptible settings can retain most attack effectiveness on smaller Whisper models.
- Introduction of a “partial suppression” objective, where the EOS token is induced within a few tokens rather than immediately, with the claim that this relaxation of the attack can further improve imperceptibility.
- Preliminary evaluation of simple audio defenses (Mu-law compression and low-pass filtering).

A potential strength of this paper is that the partial suppression formulation is a conceptual extension beyond simply re-tuning hyperparameters, and the paper attempts to test transferability and defenses for this attack model.

However, the work has several major weaknesses:
- The motivation is weak and not convincingly tied to a realistic threat model or application scenario for adversarial attacks on ASR. Much of the paper amounts to 1D hyper-parameter sweeps of an existing method, with little analysis of _why_ results occur or what general lessons can be drawn. Adversarial research is usually grounded in one of two motivations: 1) practical threats and showing that a widely deployed system can be manipulated in ways that present security, safety, or fairness risks, or 2) scientific insight: probing model robustness in order to better understand model generalization, inductive biases, or to design defenses that improve reliability. Unfortunately, this work falls into neither category convincingly and insufficient discussion is provided as to potential applications. Additionally, the paper is very short overall, with thin related work (only four total references) and minimal discussion or explanation.
- The claim of improved “imperceptibility” is not substantiated by any objective perceptual metric (e.g., PESQ, STOI) or human listening study.
- The experiments are limited to the tiny.en and small.en Whisper models on a very small subset of TED-LIUM, with only 500 training examples and single-seed runs. This makes the findings fragile and hard to generalize. Additionally, as the authors report in Section 5, the generated attacks are not transferrable across models, which further limits their practical relevance. The findings here suggest that each Whisper variant would require its own tailored attack, making deployment in real-world scenarios even less plausible and diminishing the general scientific insight that can be drawn.
- The discussion of related work is insufficient. There is a substantial literature on audio adversarial examples and defenses (e.g., Carlini & Wagner 2018; Qin et al. 2019; Yakura & Sakuma 2019; Schönherr et al. 2019; Abdullah et al. 2021) that is not cited or discussed.

**Audience:**

Yes

**Audience Explanation:**

Yes, marginally. Researchers in adversarial machine learning for speech could have some interest in the presented alternative suppression objective, but the narrow focus on Whisper, lack of clear threat model, and limited evaluation scope make it relevant to only a very small subset of readers. The paper would be more engaging to a wider TMLR audience if the authors more clearly articulated a realistic threat model, such as how adversarial ASR suppression could matter in security-critical or privacy-related contexts, and if the defense exploration were developed into a more systematic study. A deeper analysis of how different defenses mitigate or fail against these attacks could yield lessons of broader value for robustness research, extending interest beyond Whisper-specific attacks.

**Broader Impact Concerns:**

The work involves creating audio adversarial examples, which could theoretically be used for malicious purposes. The authors should consider acknowledging this risk and discussing any implications relevant to their work.

**Claims And Evidence:**

No

**Claims Explanation:**

The main claims are that: (1) existing Whisper attacks can be tuned to be less perceptible while retaining most effectiveness, and (2) partial suppression can further improve imperceptibility. While the hyperparameter sweeps support the first claim in a narrow sense, the second claim is not convincingly demonstrated — imperceptibility is never measured, and the WER/BLEU results alone are not a proxy for perceptual quality. The experiments are also limited in scope, on small models and datasets, without statistical significance or robustness analysis. The narrative is minimal, and there is a gap between the strong wording in the abstract and the evidence presented.

**Requested Changes:**

Critical for acceptance:
- Substantially strengthen the motivation and threat model. Clarify who might mount such attacks, in what scenarios, and why partial suppression is valuable.
- Provide objective or subjective evaluation of “imperceptibility” to support the main claim (e.g., PESQ/STOI scores, or a human listening study).
- Greatly expand related work, including prior ASR adversarial attacks and defenses beyond Whisper as mentioned above. Discuss how this work fits into and differs from them.
- Improve experimental rigor: evaluate on at least one larger Whisper model; increase dataset size and number of seeds to ensure reproducibility; consider evaluating on additional ASR datasets for generalization.
- Deepen the analysis: explain why certain hyperparameters work better, and what can be learned beyond Whisper.

Would strengthen the work:
- Include an overview/motivation figure early in the paper to clearly show the attack setup.
- Improve writing quality and expand further on the model setups, data preprocessing, and discussion/analysis of the results.
- Expand the defense section to include more representative and recent approaches (e.g., randomized smoothing, adversarial training, etc.).
- Discuss limitations more frankly, including the narrow model/data scope and potential lack of real-world deployment feasibility.

---

### Review · Reviewer_R7fV · 2025-08-20

**Summary Of Contributions:**

The author(s) of this paper suppression attacks on whisper models via adversarial attack. First, the author(s) show that the existing attack, which prepends a short adversarial audio patch to the audio, is effective at lower epsilon budgets, lengths than previously thought. Then, the author(s) propose a new attack, which focuses on *partial* suppression, rather than full suppression. In this new attack, the paper shows that the attack can be even more successful than the original full suppression attack at low budgets and lengths. Finally, the authors also evaluate their attack’s transferability and robustness to defenses. Results show that while their attack is not transferable, it exhibits good robustness to two defenses, but that low pass filtering has promise.

**Additional Comments:**

**Strengths**

- interesting attack (whisper model suppression)
- comprehensive hyperparameter search for existing attack limits, and initial results for potential defenses

**Weaknesses**

- the baseline comparison to “random attack snippets” is weak, as we don’t expected “random attack snippets” to work.
- while authors introduce relevant metrics to measure performance, the presentation of these metrics is confusing
- proposed defenses are limited

**Audience:**

Yes

**Audience Explanation:**

Given the limited amount of prior literature on adversarial attacks against audio transcription models, the work in this paper is timely and well appreciated. The claims are intuitive and backed up by detailed experimental results, and would definitely be useful for future studies in this subarea.

**Claims And Evidence:**

Yes

**Claims Explanation:**

This paper presents clear evidence corroborated by existing work that adversarial audio patches can effectively disabled whisper transcription models. The initial exploration of hyperparameters is comprehensive, and the results are intuitive. The newly introduced partial suppression objective and its efficacy is also easily understood and supported w/ evidence (particularily the newly introduced BLEU and WER metrics).

**Requested Changes:**

- Can the authors compare the original Raina et al attack to the no-attack baseline rather than random snippets? It’s clear that random snippets don’t work, and would be good to compare attack performance to the cleanest version of whisper transcription.
- I assume the baseline (magenta and pink lines) for Figure 1, however it’s not clear that the attack success rate (left plot) is at y = 0. Can that be made more clear? Additionally, the epsilon = 1 and epsilon = 2 are not clear to me. Are these supposed to be predetermined clamp limits? If so, doesn’t the blue and red line already cover them?
- It is difficult to compare bleu Empty/BLEU success rates and ASL/WER success rates since they are all of different scales and directions (for 2, lower means better attack, and higher means better attack for the other 2). I suggest the following:
    - Intuitively, we want high metric to = “attack success rate is high”. Can the BLEU/ASL metrics to reflect that?. I.e., 1 - current BLEU score.
    - Second, your text references figures from section 3, and claims that the attack in section 4 can achieve attack success with a lower epsilon budget. Please combine these plots so that the reader can make a direct comparison between the original attack and new attack.
- The alpha metric in 6.2 is overly complicated. I think the results would be more clear if the authors simply reported the same metrics comparing the attack with and without countermeasures in a table. The current metric misses some granularity, and only indicates if attack success changes relatively*.*
- I appreciate the author(s) evaluating the attack after performing defenses. However, I would like to know if simpler defenses such as mp3 compression are effective. Furthermore, what happens if the audio is sped up / slowed? Can these results be added?

---

### Review · Reviewer_o9Zs · 2025-10-06

**Summary Of Contributions:**

The paper investigates adversarial attacks on OpenAI’s Whisper ASR models (tiny.en and small.en). The paper focuses on short and imperceptible audio snippets that can be prepended to any audio input to partially or completely suppress the transcription capability of these models. The 3 main contributions of the paper are:

1. The authors vary attack hyperparameters like magnitude clamp limits, attack snippet length and insertion position. Through these ablations, the authors find parameters to reduce attack perceptibility from the baseline reference paper.
2. The authors modify the optimization objective to focus on partial suppression. The authors extend the above ablations to focus on partial suppression, and further improve the imperceptibility and quality of the attacks.
3. Exploration of defence methods against these attacks, particularly butterworth low pass filtering which they found to significantly reduce attack effectiveness.

Strengths:

1. Attempted extension of the referenced paper by Raina et al 2024, to partial suppression.
2. Exploration of defense strategies, wherein the authors found butterworth low pass filtering to improve metrics by around 60-70% depending on the metric.
3. Experiments to show that the attack clip learnt on small.en does not transfer to tiny.en and vice versa is good to know.

Weaknesses:
1. To be fair, these weaknesses are mentioned by the authors themselves, maybe due to lack of compute, but the experiments were performed on the 2 smallest whisper model sizes, and only one dataset was used.
2. As such it is hard to argue about the universality of these attacks (in terms of domain transfer) with these limited ablations. The cited work by Raina et al which tests transfer of the attacks between different domains and even tasks has a much stronger claim about the universality of the attacks.
3. Related to the previous point, but I do not know how useful it is to take the complete suppression work of Raina et al and optimise its hyperparameters, while at the same time severely limiting the scope of the experiments to just one dataset. Reducing the attack clip length by 0.24s might be interesting, but does this hurt the transferability of the attack to different domains?
4. Again, as mentioned by the authors themselves in section 7: Limitations, the experiments are performed only once, without any error bars, which makes drawing conclusions a stretch.
5. The novel part of the contribution, i.e. modification of the optimization objective for partial suppression presents non converged results, which again to the authors credit, they mention in the limitations section.
6. Literature survey is quite lacking, given the fact that there is quite a lot of research in this area of both finding attack vectors and defences. The paper only cites 4 other references.
7. The interesting part of the paper i.e. Section 6: Defences is too brief. Details are lacking about the butterworth low-pass filter which seems to counter these attacks. Experimental results are also quite incomplete.

**Audience:**

Yes

**Audience Explanation:**

The interesting findings of the paper revolve around partial suppression and defences, both of which are interesting, but in my opinion, not quite complete. More work is required in both the sections.

**Broader Impact Concerns:**

It is important to study such attacks on ASR systems and concurrently defences against such attacks as well.

**Claims And Evidence:**

No

**Claims Explanation:**

I belive 2 / 3 claims are not backed by convincing and clear evidence.

1. Claims of finding attack snippets that are “more imperceptible" to the human ear:. I am not sure if reducing the attack length from 0.64s to 0.4s, and clamp limit from 0.02 to 0.005 does make the attack more imperceptible. Raina et al already claims that a clamp of 0.02 is already below typical human speech signals. Does going below 0.02 make the attack more imperceptible? Maybe human listening studies would make the claim stronger.

2. Claim that attack on partial suppression is better than complete suppression attack: It would have been better to superimpose the plots for partial and complete suppression, for each of the models separately. Wrt figure 4, What is epsilon_2 and epsilon_3? How does the figure suggest a possible further decrease in clamp limits, as claimed? It is hard to compare the results from figure 4 to those of complete suppression. While the metrics for the small model do degrade significantly, the degradation is not in line with the results of complete suppression. For e.g. in Figure 2, ASL for both models hits 0, whereas in Figure 4, the small model still has minimum ASL of above 100.

3. Low pass filter seems to possess defensive power against these attacks: yes, section 6.4 does point to this.

**Requested Changes:**

1. Broader set of experiments i.e. more model sizes (if possible), more diverse datasets, more trial runs.
2. Better literature review.
3. Figures could be annotated more clearly. What is baseline small, baseline tiny, epsilon 1, 2, 3. While these can be likely inferred, it is better to be explicit.
4. Section 3 does not add a lot of value. IMHO the valuable parts of this paper are partial suppression and defences.
5. More work on partial suppression, complete the model to convergence. Superimpose plots on partial and complete suppression before drawing conclusions.
6. More work on defences. Do these defences work against partial suppression attacks. If possible to extend, do these defences work on other kinds of attacks?
7. Section 6: Defences, only focuses on the tiny whisper model. It would have been interesting to compare how the low-pass filter compares to the small model at least.
8. It would also have been interesting to compare the baseline attack with the more imperceptible attack that the authors claim to have found, and compare how the low-pass filter responds to both attacks
9. How this defence applies to partial suppression attacks would also be interesting to study.

---

### Note · Authors · 2025-10-06

I have read and agree with the venue's withdrawal policy on behalf of myself and my co-authors.